# Syntactic Question Abstraction and Retrieval for Data-Scarce Semantic Parsing

**Wonseok Hwang**                    WONSEOK.HWANG@NAVERCORP.COM
**Jinyeong Yim**                        JINYEONG.YIM@NAVERCORP.COM
**Seunghyun Park**                      SEUNG.PARK@NAVERCORP.COM
**Minjoon Seo**                         MINJOON.SEO@NAVERCORP.COM
*Clova AI, NAVER Corp.*

## Abstract

Deep learning approaches to semantic parsing require a large amount of labeled data, but annotating complex logical forms is costly. Here, we propose SYNTACTIC QUESTION ABSTRACTION & RETRIEVAL (SQAR), a method to build a neural semantic parser that translates a natural language (NL) query to a SQL logical form (LF) with less than 1,000 annotated examples. SQAR first retrieves a logical pattern from the train data by computing the similarity between NL queries and then grounds a lexical information on the retrieved pattern in order to generate the final LF. We validate SQAR by training models using various small subsets of WikiSQL train data achieving up to 4.9% higher LF accuracy compared to the previous state-of-the-art models on WikiSQL test set. We also show that by using query-similarity to retrieve logical pattern, SQAR can leverage a paraphrasing dataset achieving up to 5.9% higher LF accuracy compared to the case where SQAR is trained by using only WikiSQL data. In contrast to a simple pattern classification approach, SQAR can generate unseen logical patterns upon the addition of new examples without re-training the model. We also discuss an ideal way to create cost efficient and robust train datasets when the data distribution can be approximated under a data-hungry setting.

## 1. Introduction

Semantic parsing is the task of translating natural language into machine-understandable formal logical forms. With the help of recent advance in deep learning technology, neural semantic parsers have achieved state-of-the-art results in many tasks [Dong and Lapata, 2016, Jia and Liang, 2016, Iyer et al., 2017b]. However, their training requires the preparation of a large amount of labeled data (questions and corresponding logical forms) which is often not scalable due to the requirement of expert knowledge necessary in writing logical forms.

Here, we develop a novel approach SYNTACTIC QUESTION ABSTRACTION & RETRIEVAL (SQAR) for semantic parsing task under data-hungry setting. The model constrains the logical form search space by retrieving logical patterns from the train set using natural language similarity with assistance of a pre-trained language model. The subsequent grounding module only needs to map the retrieved pattern to the final logical form.

We evaluate SQAR on various subsets of WikiSQL train data [Zhong et al., 2017] consisting of 850∼2750 samples which occupies 1.5–4.9% of the full train data. SQAR shows up to 4.9% higher logical form accuracy compared to the previous best open sourced model SQLOVA [Hwang et al., 2019]. Also, we show that natural language sentence similarity dataset can be leveraged in SQAR by pre-training the backbone of SQAR using Quora pharaphrasing data which results in up to 5.9% higher logical form accuracy.

In general, the retrieval approach causes the limitation on dealing with unseen logical patterns. In contrast, we show that SQAR can generate unseen logical patterns by collecting new examples without re-training opening an interesting possibility of generalizable retrieval-based semantic parser.

Our contributions are summarized as follows:

- Compared to the previous best open-sourced model [Hwang et al., 2019], SQAR achieves the state-of-the-art performance on the WikiSQL test data under data-scarce environment.

- We show that SQAR can leverage natural language query similarity datasets to improve logical form generation accuracy.

- We show that retrieval-based parser can handle unseen new logical patterns on the fly without re-training.

- For the maximum cost-effectiveness, we find that it is important to carefully design the train data distribution, not merely following the (approximated) data distribution.

## 2. Related work

WikiSQL [Zhong et al., 2017] is a large semantic parsing dataset consisting of 80,654 natural language utterances and corresponding SQL annotations. Its massive size has invoked the development of many neural semantic parsing models [Xu et al., 2017, Yu et al., 2018, Dong and Lapata, 2018, Wang et al., 2017, 2018, McCann et al., 2018, Shi et al., 2018, Yin and Neubig, 2018, Xiong and Sun, 2018, Hwang et al., 2019, He et al., 2019]. Berant and Liang [Berant and Liang, 2014] built the semantic parser that uses the query similarity between an input question and paraphrased canonical natural language representations generated from candidate logical forms. In our study, candidate logical forms and corresponding canonical forms do not need to be generated as input questions are directly compared to the questions in the training data, circumventing the burden of full logical form generation. Dong and Lapata [Dong and Lapata, 2018] developed the two step approach for logical form generation, similar to SQAR using sketch representation as intermediate logical forms. In SQAR, intermediate logical forms are retrieved from train set using question similarity being specialized for data-hungry setting. Finegan-Dollak et al. [Finegan-Dollak et al., 2018] developed the model that first finds corresponding logical pattern and fills the slots in the template. While their work resembles SQAR, there is a fundamental difference between two approaches. The model from [Finegan-Dollak et al., 2018] *classifies* input query into logical pattern whereas we use query-to-query similarity to *retrieve* logical pattern non-parametrically. By retrieving logical pattern using the similarity in natural language space, paraphrasing datasets can be employed during training which is relatively easy to label compared to semantic parsing datasets. Also, in contrast to classification methods, SQAR can handle unseen logical patterns by including new examples into the train set *without* re-training the model during inference stage (see section. 5.5). Also our focus is developing competent model with small amount of data which has not been studied in [Finegan-Dollak et al., 2018]. Hwang et al. [Hwang et al., 2019] developed SQLOVA that achieves state-of-the-arts result in the WikiSQL task. SQLOVA consits of table-aware BERT encoder and NL2SQL module that generate SQL queries via slot-filling approach.

**Table**

| | # | Player | Country | Score | To par | Points | Winnings ($) |
|---|---|---|---|---|---|---|---|
| 0 | 1 | Steve Stricker | United States | 67–67–65–69=268 | –16 | 9000 | 1260000 |
| 1 | 2 | K.J. Choi | South Korea | 64–66–70–70=270 | –14 | 5400 | 756000 |
| 2 | 3 | Rory Sabbatini | South Africa | 63–71–69–68=271 | –13 | 3400 | 476000 |
| 3 | T4 | Mark Calcavecchia | United States | 67–75–65–65=272 | –12 | 2067 | 289333 |
| 4 | T4 | Ernie Els | South Africa | 65–71–68–68=272 | –12 | 2067 | 289333 |

*Q*: What is the points of South Korea player?
*L*: select Points where Country = South Korea
*l*: select #1 where #2 = #3
**Answer**: 5400

Figure 1: Example of WikiSQL semantic parsing task. For given question ($Q$) and table headers, the model generates corresponding SQL query ($L$) and retrieves the answer from the table.

## 3. Model

The model generates the logical form $L$ (SQL query) for a given NL query $Q$ and its corresponding table headers $H$ (Fig. 1). First, the logical pattern $l$ is retrieved from the train set by finding the most similar NL query with $Q$. For example in Fig. 1, $Q$ is "What is the points of South Korea player?". To generate logical form $L$, SQAR retrieves logical pattern $l$ = SELECT #1 WHERE #2 = #3 by finding the most similar NL query from the train set, for instance ["Which fruit has yellow color?", SELECT Fruit WHERE Color = Yellow]. Then #1, #2, and #3 in $l$ are grounded to Point, Country, and South Korea respectively by the grounding module using information from $Q$ and table headers. The process is depicted schematically in Fig. 2a. The detail of each step is explained below.

### 3.1 Syntactic Question Abstractor

The syntactic question abstractor generates two vector representation $q$ and $g$ of an input NL query $Q$ (Fig. 2b). $q$ is trained to represent syntactic information of $Q$ and used in the retriever module (Fig. 2c). $g$ is trained to represent lexical information of $Q$ by being used in the grounder (Fig. 2d).

The logical patterns of the WikiSQL dataset consist of combination of six aggregation operators (none, max, min, count, sum, and avg), and three where operators (=, >, and <). The number of conditions in where clause is ranging from 0 to 4. Each condition is combined by and unit. In total, there are 210 possible SQL patterns (6 select clause patterns × 35 where clause patterns, see Fig. A1). To extract these syntactic information from NL query, both an input NL query $Q$ and the queries in train set $\{Q_{t,i}\}$ are mapped to a vector space (represented by $q$ and $\{q_{t,i}\}$, respectively) via table-aware BERT encoder [Devlin et al., 2018, Hwang et al., 2019] (Fig. 2b). The input of the encoder consists of following tokens:

[CLS], $E$, [SEP], $Q$, [SEP], $H$, [SEP]

where $E$ stands for SQL language element tokens such as [SELECT], [MAX], [COL], $\cdots$) separated by [SEP] (a special token in BERT), $Q$ represents question tokens, and $H$ denotes the tokens

of table headers in which each header is separated by `[SEP]`. $E$ is included to contextualize and use them during grounding process (section 3.3). Segment ids are used to distinguish $E$ (id = 0) from $Q$ (id = 1) and $H$ (id = 1) as in BERT [Devlin et al., 2018]. Next, two vectors $q \equiv v_{0:d_q}^{[\text{CLS}]}$ and $g \equiv v_{d_q:(d_q+2d_h)}^{[\text{CLS}]}$ are extracted from the (linearly projected) encoding vector of `[CLS]` token where $i : j$ notation indicates the elements of vector between $i$th and $j$th indices. In this study, $d_q = 256$ and $d_h = 100$.

## 3.2 Retriever

To retrieve logical pattern of $Q$, the questions from the train set ($\{Q_{t,i}\}$) are also mapped to the vector space ($\{q_{t,i}\}$) using the syntactic question abstractor. Next, the logical pattern is found by measuring Euclidean $L_2$ distance between $q$ and $\{q_{t,i}\}$.

$$q_{t,i^*} = \underset{q_{t,i}}{\operatorname{argmin}} ||q - q_{t,i}||_{L_2} \tag{1}$$

Since $q_{t,i^*}$ has corresponding $Q_{t,i^*}$ and logical form $L_{t,i^*}$, the logical pattern $l^*$ can be obtained from $L_{t,i^*}$ after delexicalization. The process is depicted in Fig. 2c. In SQAR, maximum 10 closest $q_{t,i^*}$ are retrieved and the most frequently appearing logical pattern is selected for the subsequent grounding process. SQAR is trained using the negative sampling method. First, one positive sample (having the same logical pattern with input query $Q$), and 5 negative samples (having different logical pattern) are randomly sampled from the train set. Then six $L_2$ distances are calculated as above and interpreted as approximate probability by using softmax function after multiplied by -1. The cross entropy function is employed for the training.

## 3.3 Grounder

To ground retrieved logical pattern $l^*$, following LSTM-based pointer network is used [Vinyals et al., 2015].

$$\begin{aligned}
D_t &= \text{LSTM}(P_{t-1}, (h_{t-1}, c_{t-1})) \\
h_0 &= g_{0:d_h} \\
c_0 &= g_{d_h:2d_h} \\
s_t(i) &= \mathcal{W}(\mathcal{W}H_i + \mathcal{W}D_t) \\
p_t(i) &= \text{softmax } s_t(i),
\end{aligned} \tag{2}$$

where $P_{t-1}$ stands for the one-hot vector (pointer to the input token) at time $t-1$, $h_{t-1}$ and $c_{t-1}$ are hidden- and cell-vectors of the LSTM decoder, $\mathcal{W}$'s denote (mutually different) affine transformations, and $p_t(i)$ is the probability of observing $i$th input token at time $t$. Here $d_h$ (=100) is the hidden dimension of the LSTM. Compared to a conventional pointer network, our grounder has three custom properties: (1) as logical pattern is already found from the retriever, the grounder does not feed the output as the next input when the input token is already present in the logical pattern whereas lexical outputs like column and `where` values are fed into the next step as an input (Fig. 2d); (2) to generate conditional values for `where` clause, the grounder infers only the beginning and the end token positions from the given question to extract the condition values for `where` clause; (3) the multiple generation of same column on `where` clause is avoided by constraining the search space. The syntactic question abstractor, the retriever, and the grounder are together named as SYNTACTIC QUESTION ABSTRACTION & RETRIEVAL (SQAR).

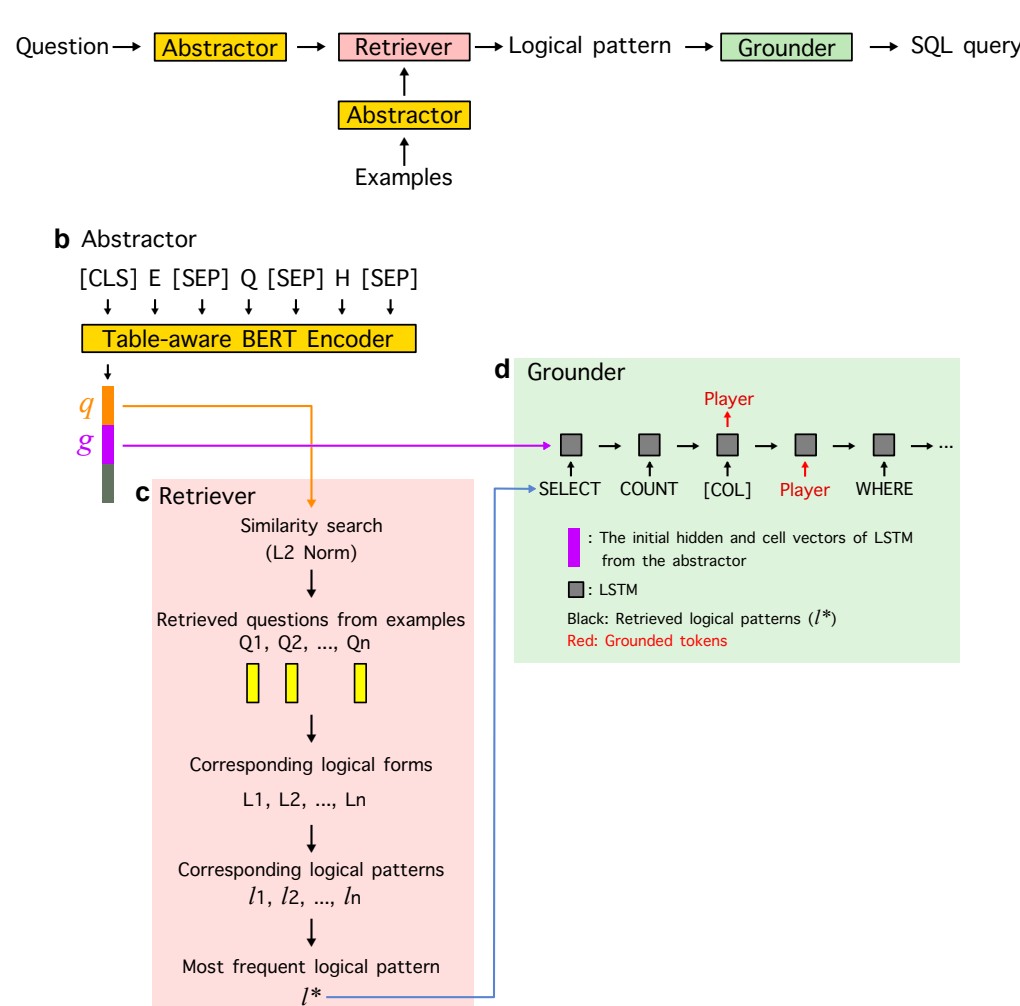

Figure 2: (a) The schematic representation of SQAR. (b) The scheme of the syntactic question abstractor. (c) The retriever. (d) The grounder. Only lexical tokens (red-colored) are predicted and used as the next input token.

## 4. Experiments

To train SQAR and SQLOVA, the pytorch version of pre-trained BERT model[1] (BERT-Base-Uncased[2]) is loaded and fine-tuned using ADAM optimizer. The NL query is first tokenized by using Standford CoreNLP [Manning et al., 2014]. Each token is further tokenized (into sub-word level) by Word-Piece tokenizer [Devlin et al., 2018, Wu et al., 2016]. FAISS [Johnson et al., 2017] is employed for the retrieval process. For the experiments with Train-Uniform-85P-850, Train-Rand-881, Train-Hybrid-85P-897, and Train-Rand-3523, only single logical pattern is retrieved from the retriever due to the scarcity of examples per pattern. Otherwise 10 logical patterns are retrieved. All experiments

---

1. https://github.com/huggingface/transformers
2. https://github.com/google-research/bert

Table 1: Comparison of models under data-hungry environment. Logical pattern accuracy (P) and full logical form accuracy (LF) on test set of WikiSQL are shown. The errors are estimated by three independent experiments with different random seeds except SQLOVA-GLOVE where the error is estimated from two independent experiments.

| Model | Train set | Dev set | P (%) | LF (%) |
|---|---|---|---|---|
| COARSE2FINE[a] | Train-Rand-881 | Dev-Rand-132 | - | $2.1 \pm 0.0$ |
| SQLOVA-GLOVE[b] | Train-Rand-881 | Dev-Rand-132 | $66.6 \pm 0.4$ | $17.6 \pm 0.3$ |
| SQLOVA[b] | Train-Rand-881 | Dev-Rand-132 | $75.3 \pm 0.4$ | $45.1 \pm 0.7$ |
| SQAR w/o Quora | Train-Rand-881 | Dev-Rand-132 | $74.1 \pm 0.8$ | $49.1 \pm 0.9$ |
| SQAR | Train-Rand-881 | Dev-Rand-132 | $75.5 \pm 0.6$ | $50.0 \pm 0.6$ |

[a] The source code is downloaded from https://github.com/donglixp/coarse2fine
[b] The source code is downloaded from https://github.com/naver/sqlova.

were performed with WikiSQL ver. 1.1 [3]. The accuracy is measured by repeating three independent experiments in each condition with different random seeds unless particularly mentioned. To further pre-train BERT-backbone of SQAR, we use Quora paraphrase detection dataset [Iyer et al., 2017a]. The further details of experiments are summarized in Appendix.

## 5. Result and Analysis

### 5.1 Preparation of data scarce environment

The WikiSQL dataset consists of 80,654 examples (56,355 in train set, 8,421 in dev set, and 15,878 in test set). The examples are not uniformly distributed over 210 possible SQL logical patterns in train, dev, and test sets while they have similar logical pattern distributions (see Fig. A1, Table 6). To mimic original pattern distribution while preparing data scarce environemnts, we prepare Train-Rand-881 by randomly sampling 881 examples from the original WikiSQL train set (1.6%). The validation set Dev-Rand-132 is prepared by the same way from the WikiSQL dev set.

### 5.2 Accuracy Measurement

SQAR retrieves SQL logical pattern for a given question $Q$ by finding most syntactically similar question from the train set and ground the retrieved logical pattern using LSTM-based grounder (Fig. 2a). The model performance is tested over the full WikiSQL test set by using two metrics: (1) logical pattern accuracy (P) and (2) logical form accuracy (LF). P is computed by ignoring difference in lexical information such as predicted columns and conditional values whereas LF is calculated by comparing full logical forms. The execution accuracy of SQL query is not compared as different logical forms can generate identical answer hindering fair comparison. Table 1 shows P and LF of several models over the WikiSQL original test set conveying following important messages: (1) SQAR outperforms SQLOVA by +4.0% in LF (3rd and 4th rows); (2) Quora pre-training improves the performance of SQAR further by 0.9% (4th and 5th rows); (3) Under data-scarce condition, the use of pre-trained language model (BERT) is critical (1st and 2nd rows vs 3–5th rows);

It is of note that COARSE2FINE [Dong and Lapata, 2018] shows much lower accuracy compared to SQLOVA-GLOVE although both models use GLoVe [Pennington et al., 2014]. One possible

---

3. https://github.com/salesforce/WikiSQL

explanation will be that COARSE2FINE first classify SQL patterns of `where` clause (sketch generation) while SQLOVA generate SQL query via slot-filling approach. The classification involves abstraction of whole sentence and this process can be a data-hungry step.

### 5.3 Generalization test I: dependency on logical pattern distribution

When the size of train set is fixed, assigning more examples to frequently appearing logical patterns (in test environment) to the train set will increase the chance for correct SQL query generation as trained model would have a higher performance for frequent patterns (Train-Rand-881 is constructed in this regard). On the other hand, including diverse patterns in train set will help the model to distinguish similar patterns. Considering these two aspects, we prepare additional two subsets Train-Uniform-85P-850, and Train-Hybrid-85P-897. Train-Uniform-85P-850 consists of 850 uniformly distributed examples over 85 patterns whereas Dev-Uniform-80P-320 consists of 320 uniformly distributed examples over 80 patterns. Train-Hybrid-85P-897 is prepared by randomly sampling examples from top most frequent 85 logical patterns. Each pattern has approximately 128 times smaller number of examples compared to the full WikiSQL train set as in Train-Rand-881. In addition, all patterns are forced to have at least 7 examples for the diversity (Fig. A1, and Table 6) resulting in total 897 examples. Only 85 patterns out of 210 patterns are considered because (1) 85 patterns occupy 98.6% of full train set, and (2) only these patterns have at least 30 corresponding examples (Fig. A1, Table 6). A dev set Dev-Hybrid-223 is constructed similarly by extracting 223 examples from the WikiSQL dev set (Fig. A1, Table 6). The difference between three types of train sets are shown schematically in Fig. 3 (orange: Train-Uniform-85P-850, purple: Train-Rand-881, black: Train-Hybrid-85P-897).

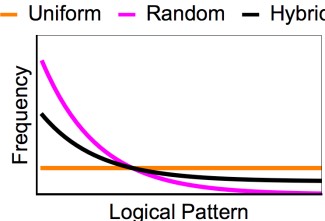

Figure 3: The schematic plot of logical pattern distribution of three types of train sets: uniform set (orange), random set (magenta), and hybrid set (black). In hybrid set, examples are distributed on logical patterns similar to random set but each logical pattern must include at least certain number of examples.

Table 2 shows following important information: (1) SQAR outperforms SQLOVA again by +4.1% LF in Train-Uniform-85P-850 (3rd and 5th rows of upper panel) and +4.0% LF in Train-Hybrid-85P-897 (3rd and 5th rows of bottom panel); (2) the Quora pre-training improves model performance +5.9% LF in Train-Uniform-85P-850 and by +0.5% LF in Train-Hybrid-85P-897(4th and 5th rows of each panel).

Both SQAR and SQLOVA show good performance when they are trained using either Train-Rand-881 or Train-Hybrid-85P-897(3rd and 5th columns of Table 1, 2). In real service delivering scenario, the data distribution in test environment could vary with time. In regard of this, we prepare

Table 2: Comparison of models with two additional train sets: Train-Uniform-85P-850 and Train-Rand-881.

| Model | Train set | Dev set | P (%) | LF (%) |
|---|---|---|---|---|
| SQLOVA | Train-Uniform-85P-850 | Dev-Uniform-80P-320 | $62.2 \pm 1.6$ | $33.8 \pm 1.2$ |
| SQAR w/o Quora | Train-Uniform-85P-850 | Dev-Uniform-80P-320 | $55.1 \pm 3.0$ | $32.0 \pm 1.7$ |
| SQAR | Train-Uniform-85P-850 | Dev-Uniform-80P-320 | $65.7 \pm 0.4$ | $37.9 \pm 0.4$ |
| SQLOVA | Train-Hybrid-85P-897 | Dev-Hybrid-223 | $77.0 \pm 0.6$ | $45.9 \pm 0.4$ |
| SQAR w/o Quora | Train-Hybrid-85P-897 | Dev-Hybrid-223 | $78.5 \pm 0.8$ | $49.4 \pm 1.2$ |
| SQAR | Train-Hybrid-85P-897 | Dev-Hybrid-223 | $78.2 \pm 0.3$ | $49.9 \pm 1.1$ |

an additional test set Test-Uniform-81P-648 by extracting 8 examples from top most frequent 81 logical patterns from the WikiSQL test. The resulting test set has completely different logical pattern distribution with the WikiSQL test set. The table 3 shows that both models show best overall performance when they are trained with Train-Hybrid-85P-897 being remained robust to the change of test environment (4th columns). The result highlights the two important properties for train set to have: reflecting test environment (more examples for frequent logical patterns), and including diverse patterns.

Table 3: Comparison of models with Test-Uniform-81P-648 having uniform pattern distribution. The numbers in the table indicates LF of two models. The model with higher score in each condition is indicated by bold face.

| Model & Test set | Train-Rand-881 | Train-Uniform-85P-850 | Train-Hybrid-85P-897 |
|---|---|---|---|
| SQLOVA, Test-Full-15878 | $45.1 \pm 0.7$ | $33.8 \pm 1.2$ | $45.9 \pm 0.4$ |
| SQLOVA, Test-Uniform-81P-648 | $\mathbf{18.9} \pm 1.5$ | $32.3 \pm 1.3$ | $31.7 \pm 1.3$ |
| SQAR, Test-Full-15878 | $\mathbf{50.0} \pm 0.6$ | $\mathbf{37.9} \pm 0.4$ | $\mathbf{49.9} \pm 1.1$ |
| SQAR, Test-Uniform-81P-648 | $17.2 \pm 1.3$ | $\mathbf{39.2} \pm 1.2$ | $\mathbf{37.6} \pm 1.7$ |

## 5.4 Generalization test II: dependency on dataset size

To further test generality of our findings under change of train set size, we prepare three additional train sets: Train-Uniform-85P-2550, Train-Rand-2677, and Train-Hybrid-96P-2750 (Table 6). Train-Uniform-85P-2550 consists of 2550 uniformly distributed examples over 85 patterns, Train-Rand-2677 consists of 2667 examples randomly sampled from the WikiSQL train data, and Train-Hybrid-96P-2750 is larger version of Train-Hybrid-85P-897 in which each logical pattern includes at least 15 examples for 96 logical patterns (Table. 6). Table 4 shows following information: (1) SQAR shows marginally better performance than SQLOVA showing +1.9%, +0.5%, and -0.7% in LF when Train-Rand-2677, Train-Uniform-85P-2550, and Train-Hybrid-96P-2750 are used as the train sets (1st and 3rd rows of each panel); (2) Again, the pre-training using Quora paraphrasing datset increases LF by +0.5%, +3.3%, and +2.7% in Train-Rand-2677, Train-Uniform-85P-2550, and Train-Hybrid-96P-2750 respectively (2nd and 3th rows of each panel); (3) Both SQAR and SQLOVA show best performance when they are trained over hybrid dataset. Observing that the performance gap between SQAR and SQLOVA becomes marginal as increasing the size of train

set, we train both models using full WikiSQL train set. The result shows that again, there is only marginal difference between two models (SQLOVA LF: 79.2 ± 0.1, SQAR LF = 78.4 ± 0.2). The overall results are summarized in Fig. 4.

Table 4: Comparison of model with three WikiSQL train subsets: Train-Rand-2677, Train-Uniform-85P-2550 and Train-Hybrid-96P-2750).

| Model | Train set | Dev set | P (%) | LF (%) |
|---|---|---|---|---|
| SQLOVA | Train-Rand-2677 | Dev-Rand-527 | 81.2 ± 0.2 | 60.9 ± 0.4 |
| SQAR w/o Quora | Train-Rand-2677 | Dev-Rand-527 | 82.0 ± 0.2 | 62.3 ± 0.5 |
| SQAR | Train-Rand-2677 | Dev-Rand-527 | 81.4 ± 0.5 | 62.8 ± 0.3 |
| SQLOVA | Train-Uniform-85P-2550 | Dev-Uniform-80P-320 | 68.2 ± 1.6 | 49.7 ± 1.2 |
| SQAR w/o Quora | Train-Uniform-85P-2550 | Dev-Uniform-80P-320 | 66.2 ± 4.5 | 47.0 ± 3.3 |
| SQAR | Train-Uniform-85P-2550 | Dev-Uniform-80P-320 | 69.0 ± 1.2 | 50.3 ± 0.7 |
| SQLOVA | Train-Hybrid-96P-2750 | Dev-Hybrid-446 | 83.1 ± 0.2 | 66.1 ± 0.6 |
| SQAR w/o Quora | Train-Hybrid-96P-2750 | Dev-Hybrid-446 | 82.2 ± 0.2 | 62.7 ± 0.2 |
| SQAR | Train-Hybrid-96P-2750 | Dev-Hybrid-446 | 82.8 ± 0.4 | 65.4 ± 1.0 |

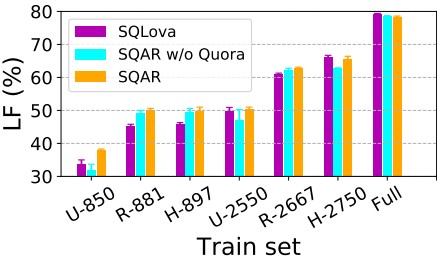

Figure 4: Logical form accuracy of two models: SQLOVA (magenta), SQAR without Quora training (cyan), and SQAR (orange) over various subsets (U-850: Train-Uniform-85P-850, R-881: Train-Rand-881, H-897: Train-Hybrid-85P-897, U-2550: Train-Uniform-85P-2550, R-2667: Train-Rand-2677, H-2750: Train-Hybrid-96P-2750, Full: the WikiSQL train set)

## 5.5 Generalization test III: parsing unseen logical forms

In general, retrieval-based approach cannot handle new type of questions when corresponding logical patterns are not presented in the train set. However, unlike simple classification approach [Finegan-Dollak et al., 2018], SQAR has interesting generalization ability originated from the use of query-to-query similarity in natural language space. The train data in SQAR has two roles: (1) supervision examples at training stage, and (2) a database to retrieve the logical pattern (a retrieval set) from which the most similar natural language query will be found during inference stage. Once the model is trained, the second role can be improved by including more examples into the train set

Table 5: Parsing unseen logical forms. SQAR is trained by using Train-Rand-881 and P and LF are measured while using a different set for query retrieval in the inference stage. R-881, H-897, H-2750, and Full stand for Train-Rand-881, Train-Hybrid-85P-897, Train-Hybrid-96P-2750, and Train-Full-56355 respectively. R-capacity indicates the number of successfully retrieved logical pattern types whereas RG-capacity indicates that of successfully parsed logical pattern types.

| Model | Train set | Set for retrieval | P (%) | LF (%) | R-capacity | RG-capacity |
|-------|-----------|-------------------|-------|--------|------------|-------------|
| SQAR | R-881 | R-881 | $75.5 \pm 0.6$ | $50.0 \pm 0.6$ | $57.5 \pm 2.2$ | $47.3 \pm 0.4$ |
| SQAR | R-881 | R-881 + H-897 | $76.6 \pm 0.4$ | $50.6 \pm 0.5$ | $79.3 \pm 1.9$ | $58.8 \pm 3.9$ |
| SQAR | R-881 | R-881 + H-2750 | $77.5 \pm 0.4$ | $50.7 \pm 0.6$ | $91.0 \pm 2.3$ | $67.0 \pm 1.7$ |
| SQAR | R-881 | Full | $79.6 \pm 0.4$ | $51.7 \pm 0.5$ | $102 \pm 2$ | $67.3 \pm 2.5$ |
| SQAR | R-881 | H-2750 | $77.2 \pm 0.5$ | $50.5 \pm 0.5$ | $92.0 \pm 2.1$ | $67.5 \pm 2.2$ |

later. Particularly, by adding examples with new logical patterns, the model can handle questions with unseen logical patterns without re-training.

To experimentally show this, we measured P and LF of SQAR while changing the retrieval set during an inference stage (Table. 5). The train set is fixed to Train-Rand-881 consisting of 67 logical patterns. The result shows that upon addition of Train-Hybrid-85P-897 into the template set, which includes 18 more logical patterns compared to Train-Rand-881, P and LF increases by 1.1% and 0.6% respectively (2nd row of the table). Similar results are observed with Train-Hybrid-96P-2750 (+2.0% in P and +0.7% LF, 3rd row of the table) and with Test-Full-15878 (+4.1% in P and +1.7% in LF, 4th row of the table). To further show the power of using query-to-query similarity, we replaced the entire retrieval set from Train-Rand-881 to Train-Hybrid-96P-2750 where only 43 examples are overlapped between them. Again, P and LF increase by 1.7% and 0.5% respectively (5th row of the table). To further confirm the addition of examples enables parsing of unseen logical patterns, we introduce two additional metrics: R-capacity and RG-capacity. R-capacity is defined by the number of successfully retrieved logical pattern types by SQAR in the test set whereas RG-capacity indicates the number of successfully generated (retrieved and grounded) logical pattern types. The table shows both R- and RG-capacities increases upon addition of examples into the retrieval set (5th and 6th columns). It should be emphasized that, during the training stage, SQAR observed only 67 logical patterns. Collectively, these results show that, SQAR can be easily generalized to handle new logical patterns by simply adding new examples without re-training. This also shows the possibility of transfer learning, even between semantic parsing tasks using different logical forms as intermediate logical patterns can be obtained from the natural language space.

## 6. Conclusion

We found that our retrieval-based model using query-to-query similarity can achieve high performance in WikiSQL semantic parsing task even when labeled data is scarce. We also found, pre-training using natural language paraphrasing data can help generation of logical forms in our query-similarity-based-retrieval approach. We also show that retrieval-based semantic parser can generate unseen logical forms during training stage. Finally, we found careful design of data distribution is necessary for optimal performance of the model under data-scarce environment.

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

## Appendix A. Appendix

### A.1 Experiments

#### A.1.1 MODEL TRAINING

To train SQAR, pre-trained BERT model (`BERT-Base-Uncased`[4]) is loaded and fine-tuned using ADAM optimizer with learning rate of $2 \times 10^{-5}$ except the grounding module where the learning rate is set to $1 \times 10^{-3}$. The decay rates of ADAM optimizer are set to $\beta_1 = 0.9, \beta_2 = 0.999$. Batch size is set to 12 for all experiment. SQLOVA is trained similarly using pre-trained BERT model (`BERT-Base-Uncased`). The learning rate is set to $1 \times 10^{-5}$ except NL2SQL layer which is trained with the learning rate $10^{-3}$. Batch size is set to 32 for all experiment.

Natural language utterance is first tokenized by using Standford CoreNLP [Manning et al., 2014]. Each token is further tokenized (into sub-word level) by WordPiece tokenizer [Devlin et al., 2018, Wu et al., 2016]. The headers of the tables and SQL vocabulary are tokenized by WordPiece tokenizer directly. FAISS [Johnson et al., 2017] is employed for the retrieval process. The PyTorch version of BERT code[5] is used. The model performance of COARSE2FINE was calculated by using the code[6] published by original authors [Dong and Lapata, 2018]. Our training of COARSE2FINE with the full WikiSQL train data results in $72 \pm 0.3$ logical form accuracy on WikiSQL test set.

All experiments were performed with WikiSQL ver. 1.1 [7]. The model performance of SQAR SQLOVA and COARSE2FINE was measured by repeating three independent experiments in each condition with different random seeds. The errors are estimated by calculating standard deviation. The performance of SQLOVA-GLOVE was measured from two independent experiment with different random seeds. For the experiments with Train-Uniform-85P-850, Train-Rand-881, Train-Hybrid-85P-897, and Train-Rand-3523, only single logical pattern is retrieved from the retriever due to the scarcity of examples per pattern. Otherwise 10 logical patterns are retrieved. The models are trained until the logical form accuracy is saturated waiting up to maximum 1000 epochs.

#### A.1.2 PRE-TRAINING WITH QUORA DATASET

To further pre-trained BERT-backbone used in SQAR, we use Quora paraphrase detection dataset [Iyer et al., 2017a]. The dataset contains more than 405,000 question pairs with a corresponding binary indicator that represents whether two questions are a pair of paraphrase or not. The task setting is analogous to the retriever of SQAR which detects the similarity of two given input NL queries and can be seen as fine-tuning in perspective of paraphrase detection task. During the training, two queries are given to the BERT model along with `[CLS]` and `[SEP]` tokens as in the original BERT training setting [Devlin et al., 2018]. The output vector of `[CLS]` token was used for the binary classification to predict whether given two queries are a paraphrase pair or not. The model was trained until the classification accuracy converges using using ADAM optimizer.

---

4. https://github.com/google-research/bert
5. https://github.com/huggingface/pytorch-pre-trained-BERT
6. https://github.com/donglixp/coarse2fine
7. https://github.com/salesforce/WikiSQL

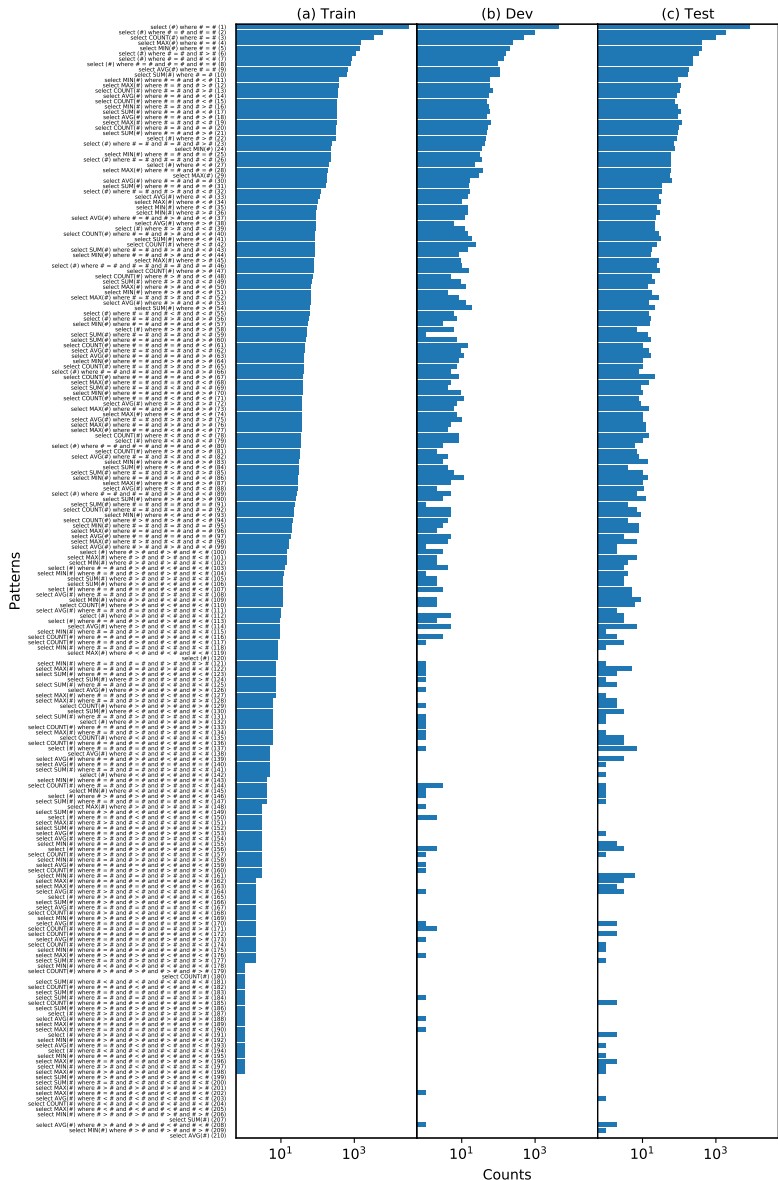

Figure A1: SQL logical patterns and their frequency in the (a) train, (b) dev, and (c) test sets of WikiSQL. The index of each pattern is represented in the parentheses on the y-axis labels.

# Appendix B. Supplementary tables

Table 6: The count of SQL logical patterns in the WikiSQL subsets used in this paper. The subset names are denoted by following shorthand notations: U-850 (Train-Uniform-85P-850), R-881 (Train-Rand-881), H-897 (Train-Hybrid-85P-897), U-2550 (Train-Uniform-85P-2550), R-2667 (Train-Rand-2677), H-2670 (Train-Hybrid-85P-2670), H-2750 (Train-Hybrid-96P-2750), UD-320 (Dev-Uniform-80P-320), RD-132 (Dev-Rand-132), HD-223 (Dev-Hybrid-223), RD-527 (Dev-Rand-527), HD-446 (Dev-Hybrid-446)

| Pattern index | Train | Dev | Test | U-850 | R-881 | H-897 | U-2550 | R-2667 | H-2670 | H-2750 | UD-320 | RD-132 | HD-223 | RD-527 | HD-446 |
|---|---|---|---|---|---|---|---|---|---|---|---|---|---|---|---|
| 1 | 30128 | 4419 | 8269 | 10 | 484 | 236 | 30 | 1425 | 471 | 942 | 4 | 84 | 35 | 305 | 70 |
| 2 | 6050 | 964 | 1798 | 10 | 95 | 48 | 30 | 263 | 95 | 190 | 4 | 10 | 8 | 52 | 16 |
| 3 | 3430 | 480 | 972 | 10 | 61 | 27 | 30 | 159 | 54 | 108 | 4 | 4 | 4 | 23 | 8 |
| 4 | 1476 | 235 | 419 | 10 | 23 | 12 | 30 | 84 | 25 | 47 | 4 | 5 | 2 | 19 | 4 |
| 5 | 1412 | 202 | 413 | 10 | 22 | 12 | 30 | 65 | 25 | 45 | 4 | 3 | 2 | 7 | 4 |
| 6 | 1116 | 158 | 339 | 10 | 14 | 9 | 30 | 53 | 25 | 35 | 4 | 3 | 2 | 15 | 4 |
| 7 | 858 | 138 | 240 | 10 | 14 | 7 | 30 | 55 | 25 | 27 | 4 | 4 | 2 | 10 | 4 |
| 8 | 794 | 92 | 229 | 10 | 13 | 7 | 30 | 30 | 25 | 25 | 4 | 1 | 2 | 3 | 4 |
| 9 | 653 | 104 | 178 | 10 | 9 | 7 | 30 | 27 | 25 | 21 | 4 | 2 | 2 | 8 | 4 |
| 10 | 619 | 105 | 173 | 10 | 11 | 7 | 30 | 30 | 25 | 20 | 4 | 0 | 2 | 3 | 4 |
| 11 | 374 | 57 | 91 | 10 | 8 | 7 | 30 | 25 | 25 | 15 | 4 | 2 | 2 | 6 | 4 |
| 12 | 369 | 59 | 107 | 10 | 5 | 7 | 30 | 23 | 25 | 15 | 4 | 0 | 2 | 5 | 4 |
| 13 | 360 | 70 | 103 | 10 | 7 | 7 | 30 | 21 | 25 | 15 | 4 | 0 | 2 | 3 | 4 |
| 14 | 357 | 55 | 88 | 10 | 9 | 7 | 30 | 17 | 25 | 15 | 4 | 1 | 2 | 2 | 4 |
| 15 | 341 | 46 | 75 | 10 | 7 | 7 | 30 | 11 | 25 | 15 | 4 | 1 | 2 | 2 | 4 |
| 16 | 334 | 55 | 92 | 10 | 5 | 7 | 30 | 17 | 25 | 15 | 4 | 0 | 2 | 0 | 4 |
| 17 | 330 | 59 | 109 | 10 | 3 | 7 | 30 | 16 | 25 | 15 | 4 | 0 | 2 | 1 | 4 |
| 18 | 329 | 48 | 89 | 10 | 3 | 7 | 30 | 19 | 25 | 15 | 4 | 0 | 2 | 1 | 4 |
| 19 | 324 | 62 | 120 | 10 | 7 | 7 | 30 | 21 | 25 | 15 | 4 | 0 | 2 | 4 | 4 |
| 20 | 319 | 50 | 97 | 10 | 2 | 7 | 30 | 9 | 25 | 15 | 4 | 1 | 2 | 5 | 4 |
| 21 | 312 | 48 | 93 | 10 | 5 | 7 | 30 | 12 | 25 | 15 | 4 | 0 | 2 | 0 | 4 |
| 22 | 309 | 45 | 83 | 10 | 2 | 7 | 30 | 15 | 25 | 15 | 4 | 0 | 2 | 0 | 4 |
| 23 | 247 | 42 | 69 | 10 | 4 | 7 | 30 | 11 | 25 | 15 | 4 | 1 | 2 | 2 | 4 |
| 24 | 234 | 35 | 76 | 10 | 3 | 7 | 30 | 10 | 25 | 15 | 4 | 1 | 2 | 1 | 4 |
| 25 | 225 | 30 | 59 | 10 | 3 | 7 | 30 | 12 | 25 | 15 | 4 | 1 | 2 | 3 | 4 |
| 26 | 222 | 35 | 59 | 10 | 2 | 7 | 30 | 9 | 25 | 15 | 4 | 0 | 2 | 0 | 4 |
| 27 | 197 | 22 | 58 | 10 | 2 | 7 | 30 | 10 | 25 | 15 | 4 | 1 | 2 | 2 | 4 |
| 28 | 188 | 36 | 59 | 10 | 3 | 7 | 30 | 7 | 25 | 15 | 4 | 0 | 2 | 0 | 4 |
| 29 | 180 | 29 | 55 | 10 | 2 | 7 | 30 | 9 | 25 | 15 | 4 | 1 | 2 | 1 | 4 |
| 30 | 179 | 16 | 61 | 10 | 2 | 7 | 30 | 12 | 25 | 15 | 4 | 0 | 2 | 0 | 4 |
| 31 | 165 | 15 | 34 | 10 | 0 | 7 | 30 | 6 | 25 | 15 | 4 | 0 | 2 | 2 | 4 |
| 32 | 123 | 16 | 33 | 10 | 1 | 7 | 30 | 6 | 25 | 15 | 4 | 0 | 2 | 2 | 4 |
| 33 | 116 | 14 | 28 | 10 | 1 | 7 | 30 | 7 | 25 | 15 | 4 | 0 | 2 | 0 | 4 |
| 34 | 100 | 10 | 31 | 10 | 1 | 7 | 30 | 3 | 25 | 15 | 4 | 0 | 2 | 0 | 4 |
| 35 | 93 | 14 | 24 | 10 | 2 | 7 | 30 | 6 | 25 | 15 | 4 | 0 | 2 | 2 | 4 |
| 36 | 91 | 14 | 29 | 10 | 2 | 7 | 30 | 4 | 25 | 15 | 4 | 0 | 2 | 1 | 4 |
| 37 | 89 | 12 | 23 | 10 | 1 | 7 | 30 | 5 | 25 | 15 | 4 | 0 | 2 | 0 | 4 |
| 38 | 89 | 6 | 21 | 10 | 0 | 7 | 30 | 4 | 25 | 15 | 4 | 0 | 2 | 0 | 4 |
| 39 | 86 | 12 | 21 | 10 | 2 | 7 | 30 | 7 | 25 | 15 | 4 | 0 | 2 | 0 | 4 |
| 40 | 84 | 14 | 28 | 10 | 2 | 7 | 30 | 6 | 25 | 15 | 4 | 1 | 2 | 1 | 4 |
| 41 | 83 | 18 | 31 | 10 | 1 | 7 | 30 | 2 | 25 | 15 | 4 | 0 | 2 | 2 | 4 |
| 42 | 82 | 24 | 25 | 10 | 3 | 7 | 30 | 3 | 25 | 15 | 4 | 0 | 2 | 1 | 4 |
| 43 | 81 | 14 | 18 | 10 | 1 | 7 | 30 | 6 | 25 | 15 | 4 | 0 | 2 | 0 | 4 |
| 44 | 81 | 8 | 17 | 10 | 2 | 7 | 30 | 3 | 25 | 15 | 4 | 1 | 2 | 1 | 4 |
| 45 | 80 | 9 | 28 | 10 | 0 | 7 | 30 | 5 | 25 | 15 | 4 | 0 | 2 | 1 | 4 |
| 46 | 77 | 10 | 25 | 10 | 1 | 7 | 30 | 3 | 25 | 15 | 4 | 2 | 2 | 2 | 4 |
| 47 | 76 | 15 | 29 | 10 | 0 | 7 | 30 | 2 | 25 | 15 | 4 | 0 | 2 | 0 | 4 |
| 48 | 74 | 5 | 18 | 10 | 0 | 7 | 30 | 3 | 25 | 15 | 4 | 0 | 2 | 0 | 4 |
| 49 | 68 | 9 | 21 | 10 | 2 | 7 | 30 | 2 | 25 | 15 | 4 | 0 | 2 | 0 | 4 |
| 50 | 66 | 13 | 14 | 10 | 1 | 7 | 30 | 3 | 25 | 15 | 4 | 0 | 2 | 0 | 4 |
| 51 | 66 | 4 | 18 | 10 | 2 | 7 | 30 | 4 | 25 | 15 | 4 | 0 | 2 | 0 | 4 |
| 52 | 66 | 8 | 27 | 10 | 0 | 7 | 30 | 5 | 25 | 15 | 4 | 0 | 2 | 0 | 4 |
| 53 | 63 | 13 | 15 | 10 | 2 | 7 | 30 | 5 | 25 | 15 | 4 | 0 | 2 | 0 | 4 |
| 54 | 62 | 19 | 21 | 10 | 2 | 7 | 30 | 4 | 25 | 15 | 4 | 0 | 2 | 1 | 4 |
| 55 | 59 | 6 | 15 | 10 | 0 | 7 | 30 | 4 | 25 | 15 | 4 | 0 | 2 | 1 | 4 |
| 56 | 56 | 7 | 17 | 10 | 1 | 7 | 30 | 1 | 25 | 15 | 4 | 0 | 2 | 0 | 4 |
| 57 | 52 | 3 | 16 | 10 | 0 | 7 | 30 | 2 | 25 | 15 | 0 | 0 | 0 | 1 | 0 |
| 58 | 51 | 6 | 7 | 10 | 1 | 7 | 30 | 1 | 25 | 15 | 4 | 0 | 2 | 0 | 4 |
| 59 | 50 | 1 | 14 | 10 | 0 | 7 | 30 | 0 | 25 | 15 | 0 | 0 | 0 | 0 | 0 |
| 60 | 46 | 7 | 17 | 10 | 0 | 7 | 30 | 2 | 25 | 15 | 4 | 0 | 2 | 0 | 4 |
| 61 | 44 | 14 | 10 | 10 | 1 | 7 | 30 | 3 | 25 | 15 | 4 | 0 | 2 | 1 | 4 |
| 62 | 44 | 9 | 15 | 10 | 0 | 7 | 30 | 3 | 25 | 15 | 4 | 0 | 2 | 2 | 4 |
| 63 | 43 | 11 | 17 | 10 | 0 | 7 | 30 | 3 | 25 | 15 | 4 | 0 | 2 | 1 | 4 |
| 64 | 43 | 8 | 10 | 10 | 1 | 7 | 30 | 4 | 25 | 15 | 4 | 0 | 2 | 0 | 4 |
| 65 | 43 | 7 | 10 | 10 | 0 | 7 | 30 | 1 | 25 | 15 | 4 | 0 | 2 | 1 | 4 |
| 66 | 41 | 5 | 8 | 10 | 0 | 7 | 30 | 2 | 25 | 15 | 4 | 0 | 2 | 2 | 4 |
| 67 | 40 | 8 | 21 | 10 | 2 | 7 | 30 | 1 | 25 | 15 | 4 | 0 | 2 | 1 | 4 |
| 68 | 39 | 5 | 15 | 10 | 0 | 7 | 30 | 3 | 25 | 15 | 4 | 0 | 2 | 0 | 4 |

| 69 | 38 | 4 | 9 | 10 | 1 | 7 | 30 | 3 | 25 | 15 | 4 | 0 | 2 | 0 | 4 |
|---|---|---|---|---|---|---|---|---|---|---|---|---|---|---|---|
| 70 | 38 | 9 | 10 | 10 | 0 | 7 | 30 | 4 | 25 | 15 | 4 | 0 | 2 | 1 | 4 |
| 71 | 36 | 11 | 8 | 10 | 0 | 7 | 30 | 1 | 25 | 15 | 4 | 0 | 2 | 3 | 4 |
| 72 | 36 | 7 | 9 | 10 | 0 | 7 | 30 | 0 | 25 | 15 | 4 | 0 | 2 | 0 | 4 |
| 73 | 36 | 6 | 15 | 10 | 0 | 7 | 30 | 1 | 25 | 15 | 4 | 2 | 2 | 2 | 4 |
| 74 | 36 | 7 | 10 | 10 | 0 | 7 | 30 | 0 | 25 | 15 | 4 | 0 | 2 | 1 | 4 |
| 75 | 36 | 10 | 10 | 10 | 0 | 7 | 30 | 2 | 25 | 15 | 4 | 0 | 2 | 2 | 4 |
| 76 | 36 | 5 | 12 | 10 | 1 | 7 | 30 | 2 | 25 | 15 | 4 | 0 | 2 | 0 | 4 |
| 77 | 36 | 4 | 12 | 10 | 0 | 7 | 30 | 1 | 25 | 15 | 4 | 0 | 2 | 0 | 4 |
| 78 | 35 | 8 | 15 | 10 | 1 | 7 | 30 | 2 | 25 | 15 | 4 | 0 | 2 | 0 | 4 |
| 79 | 35 | 8 | 10 | 10 | 1 | 7 | 30 | 2 | 25 | 15 | 4 | 0 | 2 | 0 | 4 |
| 80 | 34 | 3 | 6 | 10 | 0 | 7 | 30 | 2 | 25 | 15 | 0 | 0 | 0 | 0 | 0 |
| 81 | 33 | 2 | 7 | 10 | 0 | 7 | 30 | 2 | 25 | 15 | 0 | 0 | 0 | 0 | 0 |
| 82 | 33 | 4 | 8 | 10 | 0 | 7 | 30 | 2 | 25 | 15 | 4 | 0 | 2 | 0 | 4 |
| 83 | 31 | 3 | 14 | 10 | 2 | 7 | 30 | 1 | 25 | 15 | 0 | 0 | 0 | 0 | 0 |
| 84 | 30 | 4 | 4 | 10 | 0 | 7 | 30 | 2 | 25 | 15 | 4 | 0 | 2 | 0 | 4 |
| 85 | 30 | 6 | 10 | 10 | 0 | 7 | 30 | 2 | 25 | 15 | 4 | 0 | 2 | 0 | 4 |
| 86 | 29 | 11 | 14 | 0 | 0 | 0 | 0 | 2 | 0 | 15 | 0 | 0 | 2 | 2 | 4 |
| 87 | 29 | 5 | 10 | 0 | 0 | 0 | 0 | 0 | 0 | 15 | 0 | 0 | 2 | 0 | 4 |
| 88 | 29 | 2 | 11 | 0 | 0 | 0 | 0 | 0 | 0 | 15 | 0 | 0 | 0 | 0 | 0 |
| 89 | 27 | 5 | 7 | 0 | 2 | 0 | 0 | 1 | 0 | 15 | 0 | 0 | 2 | 0 | 4 |
| 90 | 26 | 3 | 12 | 0 | 0 | 0 | 0 | 0 | 0 | 15 | 0 | 0 | 0 | 0 | 0 |
| 91 | 24 | 1 | 4 | 0 | 0 | 0 | 0 | 0 | 0 | 15 | 0 | 0 | 0 | 0 | 0 |
| 92 | 22 | 5 | 7 | 0 | 1 | 0 | 0 | 1 | 0 | 15 | 0 | 0 | 2 | 0 | 4 |
| 93 | 22 | 5 | 9 | 0 | 0 | 0 | 0 | 2 | 0 | 15 | 0 | 0 | 2 | 0 | 4 |
| 94 | 21 | 4 | 4 | 0 | 0 | 0 | 0 | 1 | 0 | 15 | 0 | 0 | 2 | 0 | 4 |
| 95 | 20 | 3 | 8 | 0 | 0 | 0 | 0 | 1 | 0 | 15 | 0 | 0 | 0 | 0 | 0 |
| 96 | 20 | 2 | 8 | 0 | 0 | 0 | 0 | 0 | 0 | 15 | 0 | 0 | 0 | 1 | 0 |
| 97 | 18 | 5 | 3 | 0 | 0 | 0 | 0 | 0 | 0 | 0 | 0 | 0 | 2 | 0 | 4 |
| 98 | 16 | 4 | 7 | 0 | 1 | 0 | 0 | 1 | 0 | 0 | 0 | 0 | 2 | 0 | 4 |
| 99 | 16 | 1 | 2 | 0 | 0 | 0 | 0 | 0 | 0 | 0 | 0 | 0 | 0 | 0 | 0 |
| 100 | 14 | 3 | 2 | 0 | 0 | 0 | 0 | 1 | 0 | 0 | 0 | 0 | 0 | 0 | 0 |
| 101 | 14 | 2 | 7 | 0 | 0 | 0 | 0 | 0 | 0 | 0 | 0 | 0 | 0 | 0 | 0 |
| 102 | 14 | 2 | 4 | 0 | 0 | 0 | 0 | 0 | 0 | 0 | 0 | 0 | 0 | 0 | 0 |
| 103 | 13 | 4 | 3 | 0 | 0 | 0 | 0 | 1 | 0 | 0 | 0 | 0 | 2 | 0 | 4 |
| 104 | 12 | 1 | 4 | 0 | 0 | 0 | 0 | 1 | 0 | 0 | 0 | 0 | 0 | 0 | 0 |
| 105 | 11 | 2 | 3 | 0 | 0 | 0 | 0 | 1 | 0 | 0 | 0 | 0 | 0 | 0 | 0 |
| 106 | 11 | 2 | 3 | 0 | 0 | 0 | 0 | 0 | 0 | 0 | 0 | 0 | 0 | 0 | 0 |
| 107 | 11 | 3 | 5 | 0 | 0 | 0 | 0 | 1 | 0 | 0 | 0 | 0 | 0 | 1 | 0 |
| 108 | 11 | 0 | 5 | 0 | 0 | 0 | 0 | 0 | 0 | 0 | 0 | 0 | 0 | 0 | 0 |
| 109 | 11 | 2 | 9 | 0 | 0 | 0 | 0 | 1 | 0 | 0 | 0 | 0 | 0 | 0 | 0 |
| 110 | 11 | 2 | 6 | 0 | 1 | 0 | 0 | 0 | 0 | 0 | 0 | 0 | 0 | 0 | 0 |
| 111 | 10 | 0 | 2 | 0 | 0 | 0 | 0 | 1 | 0 | 0 | 0 | 0 | 0 | 0 | 0 |
| 112 | 10 | 5 | 3 | 0 | 0 | 0 | 0 | 1 | 0 | 0 | 0 | 0 | 2 | 0 | 4 |
| 113 | 9 | 2 | 3 | 0 | 0 | 0 | 0 | 0 | 0 | 0 | 0 | 0 | 0 | 0 | 0 |
| 114 | 9 | 5 | 7 | 0 | 0 | 0 | 0 | 0 | 0 | 0 | 0 | 0 | 2 | 1 | 4 |
| 115 | 9 | 0 | 1 | 0 | 0 | 0 | 0 | 0 | 0 | 0 | 0 | 0 | 0 | 0 | 0 |
| 116 | 9 | 3 | 2 | 0 | 0 | 0 | 0 | 0 | 0 | 0 | 0 | 0 | 0 | 0 | 0 |
| 117 | 8 | 1 | 3 | 0 | 0 | 0 | 0 | 0 | 0 | 0 | 0 | 0 | 0 | 0 | 0 |
| 118 | 8 | 0 | 1 | 0 | 0 | 0 | 0 | 0 | 0 | 0 | 0 | 0 | 0 | 0 | 0 |
| 119 | 8 | 0 | 0 | 0 | 0 | 0 | 0 | 1 | 0 | 0 | 0 | 0 | 0 | 0 | 0 |
| 120 | 8 | 0 | 0 | 0 | 0 | 0 | 0 | 0 | 0 | 0 | 0 | 0 | 0 | 0 | 0 |
| 121 | 7 | 1 | 1 | 0 | 0 | 0 | 0 | 1 | 0 | 0 | 0 | 0 | 0 | 0 | 0 |
| 122 | 7 | 1 | 5 | 0 | 0 | 0 | 0 | 0 | 0 | 0 | 0 | 0 | 0 | 0 | 0 |
| 123 | 7 | 1 | 2 | 0 | 0 | 0 | 0 | 2 | 0 | 0 | 0 | 0 | 0 | 0 | 0 |
| 124 | 7 | 1 | 1 | 0 | 0 | 0 | 0 | 1 | 0 | 0 | 0 | 0 | 0 | 0 | 0 |
| 125 | 7 | 0 | 2 | 0 | 0 | 0 | 0 | 1 | 0 | 0 | 0 | 0 | 0 | 0 | 0 |
| 126 | 7 | 1 | 0 | 0 | 1 | 0 | 0 | 1 | 0 | 0 | 0 | 0 | 0 | 0 | 0 |
| 127 | 7 | 0 | 1 | 0 | 0 | 0 | 0 | 2 | 0 | 0 | 0 | 0 | 0 | 0 | 0 |
| 128 | 6 | 0 | 2 | 0 | 0 | 0 | 0 | 1 | 0 | 0 | 0 | 0 | 0 | 0 | 0 |
| 129 | 6 | 1 | 2 | 0 | 0 | 0 | 0 | 1 | 0 | 0 | 0 | 0 | 0 | 0 | 0 |
| 130 | 6 | 0 | 3 | 0 | 0 | 0 | 0 | 0 | 0 | 0 | 0 | 0 | 0 | 0 | 0 |
| 131 | 6 | 1 | 1 | 0 | 0 | 0 | 0 | 1 | 0 | 0 | 0 | 0 | 0 | 0 | 0 |
| 132 | 6 | 1 | 1 | 0 | 1 | 0 | 0 | 0 | 0 | 0 | 0 | 0 | 0 | 0 | 0 |
| 133 | 6 | 1 | 0 | 0 | 0 | 0 | 0 | 0 | 0 | 0 | 0 | 0 | 0 | 0 | 0 |
| 134 | 6 | 1 | 1 | 0 | 1 | 0 | 0 | 0 | 0 | 0 | 0 | 0 | 0 | 1 | 0 |
| 135 | 6 | 1 | 3 | 0 | 0 | 0 | 0 | 0 | 0 | 0 | 0 | 0 | 0 | 0 | 0 |
| 136 | 6 | 0 | 3 | 0 | 0 | 0 | 0 | 1 | 0 | 0 | 0 | 0 | 0 | 0 | 0 |
| 137 | 5 | 1 | 7 | 0 | 0 | 0 | 0 | 0 | 0 | 0 | 0 | 0 | 0 | 0 | 0 |
| 138 | 5 | 0 | 0 | 0 | 0 | 0 | 0 | 1 | 0 | 0 | 0 | 0 | 0 | 0 | 0 |
| 139 | 5 | 0 | 3 | 0 | 0 | 0 | 0 | 0 | 0 | 0 | 0 | 0 | 0 | 0 | 0 |
| 140 | 5 | 0 | 1 | 0 | 0 | 0 | 0 | 0 | 0 | 0 | 0 | 0 | 0 | 0 | 0 |
| 141 | 5 | 0 | 0 | 0 | 1 | 0 | 0 | 0 | 0 | 0 | 0 | 0 | 0 | 0 | 0 |
| 142 | 5 | 0 | 1 | 0 | 0 | 0 | 0 | 0 | 0 | 0 | 0 | 0 | 0 | 0 | 0 |
| 143 | 4 | 0 | 0 | 0 | 0 | 0 | 0 | 0 | 0 | 0 | 0 | 0 | 0 | 0 | 0 |
| 144 | 4 | 3 | 1 | 0 | 0 | 0 | 0 | 1 | 0 | 0 | 0 | 0 | 0 | 1 | 0 |
| 145 | 4 | 1 | 1 | 0 | 0 | 0 | 0 | 0 | 0 | 0 | 0 | 0 | 0 | 0 | 0 |
| 146 | 4 | 1 | 1 | 0 | 0 | 0 | 0 | 0 | 0 | 0 | 0 | 0 | 0 | 0 | 0 |
| 147 | 4 | 0 | 1 | 0 | 0 | 0 | 0 | 0 | 0 | 0 | 0 | 0 | 0 | 0 | 0 |
| 148 | 3 | 1 | 0 | 0 | 0 | 0 | 0 | 0 | 0 | 0 | 0 | 0 | 0 | 0 | 0 |

| | | | | | | | | | | | | | | |
|---|---|---|---|---|---|---|---|---|---|---|---|---|---|---|
| 149 | 3 | 0 | 0 | 0 | 0 | 0 | 0 | 0 | 0 | 0 | 0 | 0 | 0 | 0 | 0 |
| 150 | 3 | 2 | 0 | 0 | 0 | 0 | 0 | 0 | 0 | 0 | 0 | 0 | 0 | 1 | 0 |
| 151 | 3 | 0 | 0 | 0 | 0 | 0 | 0 | 0 | 0 | 0 | 0 | 0 | 0 | 0 | 0 |
| 152 | 3 | 0 | 0 | 0 | 0 | 0 | 0 | 0 | 0 | 0 | 0 | 0 | 0 | 0 | 0 |
| 153 | 3 | 0 | 1 | 0 | 0 | 0 | 0 | 0 | 0 | 0 | 0 | 0 | 0 | 0 | 0 |
| 154 | 3 | 0 | 0 | 0 | 0 | 0 | 0 | 0 | 0 | 0 | 0 | 0 | 0 | 0 | 0 |
| 155 | 3 | 0 | 2 | 0 | 0 | 0 | 0 | 0 | 0 | 0 | 0 | 0 | 0 | 0 | 0 |
| 156 | 3 | 2 | 3 | 0 | 0 | 0 | 0 | 1 | 0 | 0 | 0 | 0 | 0 | 0 | 0 |
| 157 | 3 | 1 | 1 | 0 | 0 | 0 | 0 | 0 | 0 | 0 | 0 | 0 | 0 | 0 | 0 |
| 158 | 3 | 0 | 0 | 0 | 0 | 0 | 0 | 0 | 0 | 0 | 0 | 0 | 0 | 0 | 0 |
| 159 | 3 | 1 | 0 | 0 | 0 | 0 | 0 | 1 | 0 | 0 | 0 | 0 | 0 | 0 | 0 |
| 160 | 3 | 1 | 0 | 0 | 0 | 0 | 0 | 1 | 0 | 0 | 0 | 0 | 0 | 0 | 0 |
| 161 | 3 | 0 | 6 | 0 | 0 | 0 | 0 | 0 | 0 | 0 | 0 | 0 | 0 | 0 | 0 |
| 162 | 2 | 0 | 3 | 0 | 0 | 0 | 0 | 0 | 0 | 0 | 0 | 0 | 0 | 0 | 0 |
| 163 | 2 | 0 | 2 | 0 | 0 | 0 | 0 | 0 | 0 | 0 | 0 | 0 | 0 | 0 | 0 |
| 164 | 2 | 1 | 3 | 0 | 0 | 0 | 0 | 0 | 0 | 0 | 0 | 0 | 0 | 1 | 0 |
| 165 | 2 | 0 | 0 | 0 | 0 | 0 | 0 | 0 | 0 | 0 | 0 | 0 | 0 | 0 | 0 |
| 166 | 2 | 0 | 0 | 0 | 0 | 0 | 0 | 0 | 0 | 0 | 0 | 0 | 0 | 0 | 0 |
| 167 | 2 | 0 | 0 | 0 | 1 | 0 | 0 | 0 | 0 | 0 | 0 | 0 | 0 | 0 | 0 |
| 168 | 2 | 0 | 0 | 0 | 0 | 0 | 0 | 0 | 0 | 0 | 0 | 0 | 0 | 0 | 0 |
| 169 | 2 | 0 | 0 | 0 | 0 | 0 | 0 | 0 | 0 | 0 | 0 | 0 | 0 | 0 | 0 |
| 170 | 2 | 1 | 2 | 0 | 0 | 0 | 0 | 0 | 0 | 0 | 0 | 0 | 0 | 0 | 0 |
| 171 | 2 | 2 | 0 | 0 | 0 | 0 | 0 | 0 | 0 | 0 | 0 | 0 | 0 | 0 | 0 |
| 172 | 2 | 0 | 2 | 0 | 0 | 0 | 0 | 0 | 0 | 0 | 0 | 0 | 0 | 0 | 0 |
| 173 | 2 | 1 | 0 | 0 | 0 | 0 | 0 | 1 | 0 | 0 | 0 | 0 | 0 | 0 | 0 |
| 174 | 2 | 0 | 1 | 0 | 0 | 0 | 0 | 0 | 0 | 0 | 0 | 0 | 0 | 0 | 0 |
| 175 | 2 | 0 | 1 | 0 | 0 | 0 | 0 | 0 | 0 | 0 | 0 | 0 | 0 | 0 | 0 |
| 176 | 2 | 1 | 0 | 0 | 0 | 0 | 0 | 0 | 0 | 0 | 0 | 0 | 0 | 0 | 0 |
| 177 | 2 | 0 | 1 | 0 | 0 | 0 | 0 | 0 | 0 | 0 | 0 | 0 | 0 | 0 | 0 |
| 178 | 1 | 0 | 0 | 0 | 0 | 0 | 0 | 0 | 0 | 0 | 0 | 0 | 0 | 0 | 0 |
| 179 | 1 | 0 | 0 | 0 | 0 | 0 | 0 | 0 | 0 | 0 | 0 | 0 | 0 | 0 | 0 |
| 180 | 1 | 0 | 0 | 0 | 0 | 0 | 0 | 0 | 0 | 0 | 0 | 0 | 0 | 0 | 0 |
| 181 | 1 | 0 | 0 | 0 | 0 | 0 | 0 | 0 | 0 | 0 | 0 | 0 | 0 | 0 | 0 |
| 182 | 1 | 0 | 0 | 0 | 0 | 0 | 0 | 0 | 0 | 0 | 0 | 0 | 0 | 0 | 0 |
| 183 | 1 | 0 | 0 | 0 | 0 | 0 | 0 | 0 | 0 | 0 | 0 | 0 | 0 | 0 | 0 |
| 184 | 1 | 1 | 0 | 0 | 0 | 0 | 0 | 0 | 0 | 0 | 0 | 0 | 0 | 0 | 0 |
| 185 | 1 | 0 | 2 | 0 | 0 | 0 | 0 | 0 | 0 | 0 | 0 | 0 | 0 | 0 | 0 |
| 186 | 1 | 0 | 0 | 0 | 0 | 0 | 0 | 0 | 0 | 0 | 0 | 0 | 0 | 0 | 0 |
| 187 | 1 | 0 | 0 | 0 | 0 | 0 | 0 | 0 | 0 | 0 | 0 | 0 | 0 | 0 | 0 |
| 188 | 1 | 1 | 0 | 0 | 0 | 0 | 0 | 0 | 0 | 0 | 0 | 0 | 0 | 0 | 0 |
| 189 | 1 | 0 | 0 | 0 | 0 | 0 | 0 | 1 | 0 | 0 | 0 | 0 | 0 | 0 | 0 |
| 190 | 1 | 1 | 0 | 0 | 0 | 0 | 0 | 0 | 0 | 0 | 0 | 0 | 0 | 0 | 0 |
| 191 | 1 | 0 | 2 | 0 | 0 | 0 | 0 | 0 | 0 | 0 | 0 | 0 | 0 | 0 | 0 |
| 192 | 1 | 0 | 0 | 0 | 0 | 0 | 0 | 0 | 0 | 0 | 0 | 0 | 0 | 0 | 0 |
| 193 | 1 | 0 | 1 | 0 | 0 | 0 | 0 | 0 | 0 | 0 | 0 | 0 | 0 | 0 | 0 |
| 194 | 1 | 0 | 0 | 0 | 0 | 0 | 0 | 0 | 0 | 0 | 0 | 0 | 0 | 0 | 0 |
| 195 | 1 | 0 | 1 | 0 | 0 | 0 | 0 | 0 | 0 | 0 | 0 | 0 | 0 | 0 | 0 |
| 196 | 1 | 0 | 2 | 0 | 0 | 0 | 0 | 0 | 0 | 0 | 0 | 0 | 0 | 0 | 0 |
| 197 | 1 | 0 | 1 | 0 | 0 | 0 | 0 | 0 | 0 | 0 | 0 | 0 | 0 | 0 | 0 |
| 198 | 1 | 0 | 1 | 0 | 0 | 0 | 0 | 0 | 0 | 0 | 0 | 0 | 0 | 0 | 0 |
| 199 | 0 | 0 | 0 | 0 | 0 | 0 | 0 | 0 | 0 | 0 | 0 | 0 | 0 | 0 | 0 |
| 200 | 0 | 0 | 0 | 0 | 0 | 0 | 0 | 0 | 0 | 0 | 0 | 0 | 0 | 0 | 0 |
| 201 | 0 | 0 | 0 | 0 | 0 | 0 | 0 | 0 | 0 | 0 | 0 | 0 | 0 | 0 | 0 |
| 202 | 0 | 1 | 0 | 0 | 0 | 0 | 0 | 0 | 0 | 0 | 0 | 0 | 0 | 0 | 0 |
| 203 | 0 | 0 | 1 | 0 | 0 | 0 | 0 | 0 | 0 | 0 | 0 | 0 | 0 | 0 | 0 |
| 204 | 0 | 0 | 0 | 0 | 0 | 0 | 0 | 0 | 0 | 0 | 0 | 0 | 0 | 0 | 0 |
| 205 | 0 | 0 | 0 | 0 | 0 | 0 | 0 | 0 | 0 | 0 | 0 | 0 | 0 | 0 | 0 |
| 206 | 0 | 0 | 0 | 0 | 0 | 0 | 0 | 0 | 0 | 0 | 0 | 0 | 0 | 0 | 0 |
| 207 | 0 | 0 | 0 | 0 | 0 | 0 | 0 | 0 | 0 | 0 | 0 | 0 | 0 | 0 | 0 |
| 208 | 0 | 1 | 2 | 0 | 0 | 0 | 0 | 0 | 0 | 0 | 0 | 0 | 0 | 0 | 0 |
| 209 | 0 | 0 | 1 | 0 | 0 | 0 | 0 | 0 | 0 | 0 | 0 | 0 | 0 | 0 | 0 |
| 210 | 0 | 0 | 0 | 0 | 0 | 0 | 0 | 0 | 0 | 0 | 0 | 0 | 0 | 0 | 0 |