# OpenReview forum: "Syntactic Question Abstraction and Retrieval for Data-Scarce Semantic Parsing"
_AKBC.ws/2020/Conference — AKBC 2020_

### Official Review · AnonReviewer2 · 2020-03-27
**Retrieval-Based Data-Scarce Semantic Parsing**

**Rating:** 8
**Confidence:** 5

**Review:**

The paper presents a retrieval-based semantic parsing method for the data-scarce setting. The model retrieves a logical pattern from the train data by computing the similarity between NL queries. Then lexicons are added to the retrieved pattern in order to generate the final LF. The motivation and the proposed method make sense to me. The experimental results also show that the approach improves the performance over several baseline methods. The only concern is that the experiments are only conducted on wikisql, which is not very data-hungry. The paper reduces the number of training set to simulate the setup. The paper could be improved by conducting experiments on more small-scale datasets.

---

> ### Author Response · Authors · 2020-04-10
> **We appreciate the reviewer for acknowledging the value of our work and providing constructive comments.**
>
> We appreciate the reviewer for acknowledging the value of our work and providing constructive comments. WikiSQL was selected as our first test bed as it provides the large volume of examples allowing us to analyze the model performance in different data scales. We will also employ other (small size) semantic parsing datasets to further solidify and develop retrieval based parsing approach under data-scarce environments in the future.

---

### Official Review · AnonReviewer1 · 2020-03-27
**Solid Paper on low data semantic parsing with deep analysis**

**Rating:** 6
**Confidence:** 3

**Review:**

Summary:  This paper introduces a modeling technique for Text-to-SQL semantic parsing designed to work well in data sparse regimes. The model works by first retrieving the most similar questions from the training set. The SQL logical form of  retrieved questions are then "ungrounded" and the most common retrieved logical form pattern is then
 fed into a grounding network which insert entities from the question into the pattern to form the final parsed logical form.

Strengths:
- The authors demonstrate improved performance over a SQLova baseline in the small data regime and comparable performance when more data is available
- The authors perform thoughtful "generalization tests" to investigate potential weaknesses or behaviors of their model (the dependence on logical pattern distribution, dependence on the dataset size and generalizing to unseen forms)
- The separation of syntactic and lexical parts of the parsing process is interesting and sensible.
- Their method is able to leverage additional question similarity resources which allow their model's performance to improve without requiring extra expensive parsing annotation.

Weaknesses:
- The parsing method is not compositional, harming its generalizability. The model can never generalize to patterns not seen in its database of patterns, but there may well be training signals in the dataset that would allow for this kind of behavior.
- Performance gains, whilst certainly present, are relatively modest over SQLova in most settings.
- The authors demonstrate that the model can generalize to patterns not seen at training time by adding extra data to the model's database at test time. Whilst this boosts performance, it seems the performance is worse than simply training the model again with the extra data.

---

> ### Author Response · Authors · 2020-04-10
> **We thank the reviewer for critical reading of our manuscript indicating the strength and weakness keenly.**
>
> We thank the reviewer for critical reading of our manuscript indicating the strength and weakness keenly. SQAR focuses on generating SQL queries with logical patterns "observed during training" with high accuracy under data scarce condition via retrieval approach that may potentially sacrifice generalization performance for unseen logical patterns. However, we found SQAR can also handle the query with unseen logical pattern during training at the inference step by including new examples to the dataset without re-training the model. Making SQAR use newly included examples more efficiently and be compositional boosting its performance further remains as our future work.

---

### Official Review · AnonReviewer3 · 2020-03-30
**Good focused contribution showing the efficacy of retrieval based models for low resource semantic parsing**

**Rating:** 7
**Confidence:** 4

**Review:**

This paper describes a retrieval-based model which uses query-to-query similarity for the WikiSQL semantic parsing task. The method especially does well when labeled data is scarce.

The approach is simple yet effective. The paper is very well written, the experiments are incisive and clearly demonstrate the usefulness of the approach.

Can we also compare this model with other supervised approaches such as Berant and Liang, and especially Finegan-Dollak et al.. This comparison will help the readers understand the value of the query similarity based non-parametric approach.

It would also be interesting to see how well this model does/fails when the queries become more and more complex and compositional. Current approach seems to work only for specific kinds of queries.

The syntactic information of the queries $q$ being used in the retriever module and the lexical representation $g$ being used in the grounder are obtained by slicing the encoding of the CLS token in the table aware BERT encoder. This is strange. What guarantees that the syntactic and semantic representation can be disentangled in this way. Please see this paper for a better way to do this: https://www.aclweb.org/anthology/N19-1254/

A small description of SQLOVA would help.

typo: 141: environments

---

> ### Author Response · Authors · 2020-04-10
> **We thank the reviewer for acknowledging the value of our work and giving productive comments.**
>
> We thank the reviewer for acknowledging the value of our work and giving productive comments. We reply to individual comments as below.
>
> [About comparison with other models]
> Finegan-Dollak et al. develop neural semantic parser that generates SQL query for given input query by selecting a logical pattern and corresponding entities from the utterance via LSTM classifier. Although the approach is similar to SQAR, retrieving logical pattern and grounding, the use of query similarity during retrieval process in SQAR has following merits: (1) by retrieving logical pattern via similarity in natural language space, paraphrasing datasets can be employed which is relatively easy to construct compared to semantic parsing datasets. Also, other non-SQL semantic parsing datasets can be employed to train SQAR. (2) SQAR can parse unseen logical patterns during training by adding new  examples without re-training.
>
> Berant and Liang developed the model that first generates candidate logical forms and corresponding canonical utterances for given input utterance. The most similar utterance with respect to the input utterance among the generated ones is selected and corresponding logical form is used as the model output. Although SQAR also uses the similarity in natural language space, it circumvents the burden of generating candidate logical form and canonical utterance by directly retrieving them from examples.
> We will update the "Related works" section of our manuscript accordingly.
>
>
> [More complex queries]
> WikiSQL was selected as our first test bed as it provides the large volume of examples facilitating the measurement of scale-dependent behaviour of SQAR and it consists of relatively simple queries. We will extend the task to parse more complex queries in the future.
>
>
> [About separation of syntactic (q-vector) and semantic (g-vector) information]
> We thank the reviewer for providing valuable reference. During training, SQAR employs two losses: (1) the loss from the retrieval process estimated by Euclidean distance between q-vectors, and (2) the loss from the grounder. q-vectors should remove semantic information to minimize the loss as the questions with same syntactic information should map to the identical vector. On the other hand, g-vector should include semantic information to properly ground logical patterns although there is no guarantee that syntactic information is completely removed from it. The additional loss like word ordering suggested in Chen et al., may be employed in the future study to improve the separation process.
>
>
> [A brief description of SQLova]
> SQLova is a neural semantic parser that generates SQL queries. First, SQLova encodes question and table headers using table-aware BERT encoder. Next, it generates SQL queries via slot-filling approach by classifying individual components: aggregation operator and corresponding columns in select clause, and the number of conditions, column and corresponding operator and value in where clause.
> We will update the "Related works" section of our manuscript accordingly.

---

### Author Response · Authors · 2020-04-10
**We thank all the reviewers for acknowledging the value of our approach under a data scarce environment and the solid generalization test.**

We thank all the reviewers for acknowledging the value of our approach under a data scarce environment and the solid generalization test. We also deeply appreciate productive comments from all reviewers giving insights about how SQAR can be improved further. In the future work, we will extend our work over compositional queries using various text-to-SQL tasks and improve SQAR to have better performance and generalization ability.

---

### Decision · Program_Chairs · 2020-05-01

**Decision:**

Accept

**Comment:**

This paper proposed a simple and effective retrieval-based approach for text-to-SQL semantic parsing for the data-scarce setting. The approach has been evaluated on the WikiSQL dataset and demonstrates gains over the previous best model SQLOVA when a small number of training examples were used. It also demonstrates a zero-shot ability to handle unseen logical patterns.

All the reviewers agreed that this paper is well-written and the approach is effective and well-justified in the experiments. Therefore, we recommend the acceptance of this paper.

A major concern raised among the reviewers is whether this approach can be extended to other truly small semantic parsing datasets and more compositional logic forms. This is worth exploring and can leave it to future work.